# Thermal Characterizations of Waste Cardboard Kraft Fibres in the Context of Their Use as a Partial Cement Substitute within Concrete Composites

**DOI:** 10.3390/ma15248964

**Published:** 2022-12-15

**Authors:** Robert Haigh, Paul Joseph, Malindu Sandanayake, Yanni Bouras, Zora Vrcelj

**Affiliations:** Institute for Sustainable Industries and Liveable Cities, Victoria University, Melbourne, VIC 3011, Australia

**Keywords:** kraft fibres, cement composites, waste cardboard, silica fume, metakaolin, thermophysical properties

## Abstract

The building and construction industry consumes a significant amount of virgin resources and minimizing the demand with alternative waste materials can provide a contemporary solution. In this study, thermal components of kraft fibres (KFs) derived from waste cardboard are investigated. The mechanical properties containing KFs within concrete composites are evaluated. Metakaolin (MK) and KFs were integrated into concrete samples as a partial substitute for cement. Silica Fume (SF) was applied to the KF (SFKFs) with a view to enhancing the fibre durability. The results indicated that there was a reduction in compressive strength of 44 and 56% when 10% raw and modified KFs were integrated, respectively. Raw, fibre and matrix-modified samples demonstrated a 35, 4 and 24% flexural strength reduction, respectively; however, the tensile strength improved by 8% when the matrix was modified using MK and SFKF. The morphology of the fibres was illustrated using a scanning electron microscope (SEM), with an energy dispersion X-ray spectroscopy (EDS) provision and Fourier transform infrared spectroscopy (FT-IR) employed to gain insights into their chemical nature. The thermal, calorimetric and combustion attributes of the fibres were measured using thermogravimetric analysis (TGA), differential scanning calorimetry (DSC) and pyrolysis combustion flow calorimetry (PCFC). SFKFs showed a lower heat release capacity (HRC), demonstrating a lower combustion propensity compared to raw KFs. Furthermore, the 45% decreased peak heat release rate (pHRR) of SFKFs highlighted the overall reduction in the fire hazards associated with these materials. TGA results also confirmed a lower mass weight loss of SFKFs at elevated temperatures, thus corroborating the results from the PCFC runs.

## 1. Introduction

In recent times, researchers have been highly focused on investigating novel sustainable construction materials due to the negative environmental impacts caused by excessive virgin material consumption [1,2,3]. Global population growth has increased the demand for housing and infrastructure [4,5]. There has been an increased demand for concrete, with worldwide production of cement increasing by more than 24% in the last decade, resulting in approximately 5% of annual global greenhouse gas emissions (GHG) [5,6]. Significant research has been conducted to reduce the volume of cement in concrete and mortar materials due to the high embodied carbon and energy in cement [7]. Several researchers have investigated using supplementary cementitious waste materials as a partial cement substitute within concrete. This has been in order to counter both issues of reducing virgin materials used in cement production and addressing excessive waste generation [8,9,10,11,12]. 

Due to the abundance of residential waste, these materials are becoming increasingly researched as a potential source of building and construction material. In recent years, there has been a strong interest towards the integration of waste cardboard within cement and mortar materials [13,14,15]. To effectively use cardboard within a cementitious composite, cardboard must be reduced to either a pulp or fibrous material [16]. The main constituent fibres that give cardboard its strength are called kraft fibres (KFs). KFs are natural cellulose fibres derived from plants and trees [17]. These fibres have advantageous properties such as ease of availability, biodegradability, low density and non-abrasiveness [18]. However, cement has a high alkaline value, which can degrade KFs significantly. Researchers have identified that supplementary cementitious materials (SCMs) can lower the alkalinity and enhance fibre to matrix integration [19,20]. Generally, fibre and matrix modifications are required to enhance the mechanical stability of KFs within a cement environment [14]. This was shown by Booya et al. [21] using silica fume (SF), ground blast furnace slag (GBFS) and metakaolin (MK) to improve the strength and permeability characteristics of fibre-reinforced concrete. Mohr et al. [19] integrated FA, GBFS, MK, SF and calcined diatomaceous earth and volcanic ash (DEVA) within fibre cement boards. Their results demonstrated that binary and ternary blends of SCMs improved the durability of the fibre composites; however, this improvement can be attributed to the significant reduction of calcium hydroxide (Ca(OH)_2_) and the stabilization of the alkali content. Research integrating KFs within cement-based composites has predominantly focused on the mechanical properties [14], with further research required to understand the thermal characteristics of raw KFs and surface-modified KFs before integration within cementitious composites. 

An important factor of natural-fibre-reinforced cementitious composites is the materials’ level of decomposition that can occur with increased temperature. Previous research [22] has reported on the thermal properties of natural fibres (NFs) within polar and non-polar composite designs. However, reports on the thermal properties of KFs used in cementitious composite designs are very limited. During the kraft-making process, fibre constituents such as lignin and hemicellulose are removed from the fibre walls [23]. Mild pyrolysis has been shown to mainly effect these elements of NFs, creating a pseudo-lignin that increases the hydrophobic properties [24]. When NF cementitious composites decompose due to increased temperatures, chemical and physical changes can occur. These changes include hydrolysis, oxidation, decarboxylation, depolymerization, debonding and dehydration [25]. Owing to the removal of those fibre elements, further investigations are required of the thermal reactivity when KFs are integrated within a cementitious matrix. The determination of thermal properties can aid towards an appropriate application of KF composites for further use in the building and construction industry. In the present study, in order to accommodate KFs within the cementitious matrix, MK was integrated as the SCM. Dehydroxylation of kaolinite and disordered kaolinite can produce metakaolin and metadiskaolin, respectively. Both types of kaolin can be supplemented for partial cement substitution [26]. These variants of kaolinite generally contain a high percentage of alumina (Al_2_O_3_), which has been shown to enhance the hydration of cement in the early stages [27]. This factor also increases the durability of NFs because of the rapid consumption of Ca(OH)_2_, which in turn reduces the alkali attack on fibre walls [19,21,28,29]. When the durability of the fibre is enhanced, the mechanical properties are less hindered by the degradation that can occur due to the high alkaline percentage. Silica fume (SF) was applied as the fibre surface modification to the raw KFs. The integration of SF can increase the mechanical and durability performance of cement composites [13,15]. It can also enhance the formation of calcium silicate hydrate (C-S-H) within the cement matrix, thus increasing the composites’ mechanical strength. SF generally contains a high percentage of silicon dioxide (SiO_2_), which can consume Ca(OH)_2_ at later stages of hydration. These factors can also increase the bond between the fibre and the matrix. 

This paper aims to evaluate the thermal performance of cardboard, cardboard pulp, and raw and SF surface-modified KFs. The mechanical strength properties of the kraft fibre concrete composite materials are also presented with a view to illustrating and exemplifying the physical attributes of the fibrous material. The main objective of the present study is to demonstrate the successful integration of cardboard waste material as a partial cement substitute within concrete composite materials for further applications within the building and construction industry. 

## 2. Experimental Procedure

### 2.1. Raw Materials

The main constituent materials used to reduce the consumption of OPC within the mix design of the concrete specimens were waste corrugated cardboard and MK. Figure 1 graphically depicts the sequential process used to reduce waste cardboard into a fibrous material that is viable for concrete integration. As illustrated, the waste cardboard was initially converted into a pulp material and then a specific type of fibre design was combined with SF to create Silica Fume Kraft Fibre (SFKF). The KFs contained an approximate 20 wt. % loading of SF. In the second type of fibre design, only the raw material without any SF attachment (KF) was employed. Moisture was then removed from both fibre designs through drying via a conventional oven at 20 °C for 8 h. The fibres were then subjected to rotation via a blender mixer. The final fibrous material was the result of this eight-step process, and can be subsequently integrated within the concrete mix design. Metakaolin was used within the composites matrix, conforming with the ATSM C-618 [30], i.e., Class N specifications for natural and calcined pozzolans. Silica fume conformed with the Australian Standard AS/NZS 3582.3 [31] specification of Silica Fume used in cementitious mixtures. Ordinary Portland Cement (OPC) was used as the primary constituent for the pozzolanic reactivity and conformed with AS/NZS 3972 [32]. Locally available coarse and fine aggregate were applied, conforming to AS/NZS 1141.6.2 [33] and AS/NZS 1141.5 [34], respectively. The composition of the pozzolanic materials is shown in Table 1. 

### 2.2. Specimen Preparation

A mortar mixer was used to prepare concrete specimens. The raw materials were dry mixed for 5 min before adding water. After adding water, the materials were mixed for an additional 5 min. Upon completion of the mixing process, concrete was poured into various moulds in three separate layers with twenty compactions for each layer using a steel rod to maintain uniform compaction. The specimens were prepared at a room temperature of 20 °C for 24 h, before curing was applied in water baths for 7, 14 and 28 days. 

### 2.3. Mix Design

Raw KFs and SFKFs were thermally analysed to evaluate their degradation characteristics. The mechanical properties of concrete when reducing OPC with SFKFs, raw KFs and MK were analysed. Three bespoke mix designs were analysed in this study, with the mix code correlating to the constituent materials that were used in the concrete. For example, the SFKF105 sample is the integration of 10% SFKFs and 5% MK, whereas, correspondingly, SFKF10 is 10% SF modified fibres and KF10 is 10% raw KFs. The total reduction of OPC within each mix design is shown in Table 2. Water, fine and coarse aggregates remained the same as the control with a mass ratio of 0.33, 1.15 and 1.73 per kilogram of concrete, respectively. The workability of the materials remained consistent during the batching process regardless of fibre integration. 

### 2.4. Testing Procedure

#### 2.4.1. Thermogravimetric Analysis (TGA)

Thermogravimetric analysis (TGA) is generally performed in order to gain insights into the thermal and thermo-oxidative behaviour of solid materials. In the present study, the TGA runs were conducted on samples (on powdered materials accounting to ca. 10–15 mg) of waste cardboard (CB), waste cardboard dried pulp (CBDP), raw KFs and SFKFs. Generally, TGA provides data relating to the material’s decomposition and can demonstrate the degradation kinetics and char formation propensities. A Mettler Toledo instrument was used in this study, in accordance with ASTM E1131 [35]. The tests were conducted on powdered samples (ca. 10–15 mg) in a nitrogen atmosphere and at two different heating rates, 10 °C min^−1^ and 60 °C min^−1^, and from 30 to 900 °C, with a gas flow rate of 50 mL min^−1^. The second heating rate of 60 °C min^−1^ was chosen with a view to directly comparing the results with those from the pyrolysis combustion flow calorimetry (PCFC) experiments.

#### 2.4.2. Differential Scanning Calorimetry (DSC)

The differential scanning calorimetry (DSC) analyses of the samples were performed with a view to identifying phase changes and measuring the energetics of the pyrolytic reactions. For this purpose, DSC tests were conducted on some of the pristine specimens, and fibrous component (KFs) and its admixture with silica fumes (SFKFs). For these measurements, a Mettler Toledo instrument was employed, with the tests conducted in an atmosphere of nitrogen and at a heating rate of 10 °C min^−1^ (from 30 to 550 °C). Accurately weighed samples (ca. 15 mg) were initially taken into aluminium crucibles, with these crucibles subsequently sealed with aluminium lids that contained pinholes, thus facilitating the escape of any gaseous products that were formed, especially at elevated temperatures.

#### 2.4.3. Fourier Transform Infrared Spectroscopy (FT-IR)

Fourier transform infrared spectroscopy (FT-IR) was used to obtain information relating to the chemical nature of the test samples. A PerkinElmer 1600 Series spectrometer was used in this study in accordance with ASTM E1252 [36]. The test specimens were irradiated with infrared rays in the range from 4000 to 600 cm^−1^ in the attenuated total reflectance (ATR) mode. In each case, a few milligrams of the test sample were used as neat and it was mounted onto the diamond crystal stage. A plot of absorbance versus wavenumber was generated for each of the samples, after appropriate baseline correction for each run (number of scans: 16; resolution: 4 cm^−1^), which aided in identifying the various functional groups present within them. 

#### 2.4.4. Pyrolysis Combustion Flow Calorimetry (PCFC)

This technique, also known as “microscale combustion calorimetry” (MCC), is a small-scale calorimetric testing method used to analyse the fire behaviour of various solid materials when subjected to a forced non-flaming combustion, under anaerobic or aerobic conditions. This is conducted in accordance with ASTM D7309 [37]. The seminal work behind this technique was performed at the Federal Aviation Administration in the USA in the late 1990s. This method is assumed to reproduce and decipher the condensed and gaseous parts of flaming combustion in a non-flaming test regime, through a rapid and controlled pyrolysis of the sample in an inert atmosphere (i.e., in nitrogen) followed by high-temperature oxidation (i.e., combustion) of the pyrolyzate components in the presence of oxygen [38].

The main advantage of using PCFC is that only a very small quantity (mg) of a sample is required, and the test method often provides information regarding useful combustion parameters, such as peak heat release rate (pHHR), temperature to pHHR, total heat released (THR), heat release capacity (HRC), effective heat of combustion (EHC) and percentage of char yield. Although PCFC can provide some of the useful data regarding heat-related parameters in the smaller scale, correlation of the available data with real fire scenarios is still limited. In the present work, PCFC runs were conducted in some chosen substrates (mainly step-growth polymers) at 1 °C min^−1^, applying an FTT microscale calorimeter using method A, i.e., in an atmosphere of nitrogen.

#### 2.4.5. Mechanical Testing

Compressive strength was determined using concrete cylinders of 100 mm diameter and 200 mm length. The load rating applied to the specimens was 20 MPa/min in accordance with AS 1012.9 [39]. The flexural strength was determined using a four-point bending test. The size of the specimens used in this experiment were 100 mm × 100 mm × 350 mm. The load rate applied to the concrete beams was 1 MPa/min in accordance with AS 1012.11 [40]. The tensile strength of the samples was determined via the indirect tensile testing method. The size of the cylindrical specimens used in this experiment were 100 mm × 200 mm. The load rating applied to the concrete cylinders was 1.5 MPa/min in accordance with AS 1012.10 [41]. The mechanical testing equipment used was the Matest C088-11N Servo-Plus evolution testing machine and Cyber-Plus evolution data acquisition system. The tensile, compressive and flexural results were measured at 7-, 14- and 28-day intervals. The average was determined from three specimen samples.

#### 2.4.6. Scanning Electron Microscopy Analysis

A scanning electron microscopy (SEM) with an energy dispersion provision (EDS) was used to analyse the morphological features of the samples. This was conducted on the Phenom XL G2 Desktop SEM operating at 10 kV via 1000 magnitude. Samples were reduced to 2–5 mm in diameter using a diamond cutting saw. The EDS report illustrated the chemical composition of the samples provided.

## 3. Results and Discussion

### 3.1. Chemical Characteristics of the Base Materials (FT-IR)

As expected, the test materials (CB, CBDP, KFs and SFKFs) exhibited specific absorptions (i.e., broad signal centred around 3250 cm^−1^ owing to -O-H stretching and -C-H stretching occurring in the region of 2900 cm^−1^), and generated a typical fingerprint region of a ligno-cellulosic material, in their respective spectrum (a representative spectrum from CBDP is shown in Figure 2). These are measured at approximately 725 cm^−1^, 1450 cm^−1^, 2800 cm^−1^ and 2900 cm^−1^. The peaks at 725 cm^−1^ are correspondent to impurities such as residual waxes and lignin. 

### 3.2. Thermal and Calorimetric Properties (TGA and DSC)

The TGA runs were conducted in an inert atmosphere of nitrogen and at two heating rates, 10 and 60 °C min^−1^, the latter with a view to replicating the heating rate that was employed for the PCFC runs (i.e., 1 °C s^−1^). It is relevant to note here that the general profiles of the thermograms for test samples were similar regardless of the heating rate and consisted of the following distinctive steps: 1. Initial loss of water (i.e., mainly desorption of the physically bound water) up to 180 °C; 2. Main chain degradation, including dehydration reactions up until about 400 °C; 3. Secondary degradation of the carbonaceous residue until the end of the run (900 °C). This is also in accordance with researchers stating that there are three main stages of weight loss with natural fibres [23,27]. Firstly, a slight weight loss is exhibited between 50–110 °C due to the evaporation of moisture in the fibre. Secondly, the decomposition of hemicellulose and primary component of lignin at temperatures between 270–360 °C. Finally, maximum weight loss occurs when the cellulose component is degraded heavily between 350–500 °C [27]. Prins et al. [23] also demonstrated that the decomposition of wood fibre occurs between 225–310 °C for hemicellulose, residual lignin between 250–500 °C and cellulose between 305–375 °C. During the manufacturing of KFs for production of cardboard materials, residual components such as lignin are mostly removed via chemical and thermal treatment. It is interesting to note that the mass loss exhibited occurred above 180°C, whereas Urrea-Ceferino et al. [42] conducted TGA experiments on unbleached northern softwood KFs and noted an abrupt mass loss between 105–150 °C. This highlights prior thermal modification during the manufacturing of KFs. The individual mass losses and corresponding temperatures are provided in Table 3, while a typical thermogram (CB at 10 °C min^−1^) is shown in Figure 3.

As shown in Table 3, the extent of water retention among the various samples differed, albeit within a relatively narrower range (between 3 and 7 wt. %); however, the corresponding values for mass losses at 480°C varied more widely. It is also interesting to note that these values also seem to be influenced by the heating rates, and this aspect is also reflected in the amounts of char residues that were left after the completion of the runs. Given the presence of the inorganic filler (SF~20 wt. %) in SFKFs, the maximum amount of char residue was exhibited at 900 °C, while the minimum extent of mass losses was recorded at 480 °C. Tonoli et al. [43] conducted an application of acid hydrolysis to whisker fibres and noted a 40% weight loss with temperatures between 150–310 °C. Further degradation was shown to occur within a temperature range of 350–525 °C; however, above 550 °C, no thermal event occurred. This demonstrates that the application of SF modification on the KF walls delays degradation at high temperatures.

As expected, the DSC curves of the samples (for example, SFKFs) showed at least two distinctive peaks which mirrored their corresponding TGA thermograms. Evidently, these represent endothermic peaks that correspond to the loss of physically bound water (between 30 and 180 °C) and main chain pyrolysis reactions (occurring between 250 and 400 °C). A broader endothermic peak beyond 400 °C (and until the end of the run: 550 °C) could be indicative of slow degradation reactions of the residue that was left after the main chain degradation (see Figure 4).

### 3.3. Combustion Attributes (PCFC)

The relevant parameters obtained through PCFC runs are tabulated in Table 4. Generally, the heat release capacity (HRC) is considered as a reliable indicator of the combustibility of a material [38]. Evidently, the HRC value was found to be the lowest for the SFKF sample (so also the corresponding values for THR, pHRR, char yields and EHC). Therefore, it can be concluded that the presence of the inorganic filler, SF, substantially aided in reducing the overall flammability of the fibrous matrix. This factor of fibre modification is critical for the durability of fibre cement composites when exposed to increased or sustained elevated temperatures. Figure 5 illustrates the microstructure of the SFKFs, demonstrating the successful application of SF on the KF walls. The EDS plot further features the attachment of SiO_2_ (SF) to the fibrous matrix. Onésippe et al. [24] noted that the inclusion of SF within cement composites can reduce the specific heat by 41%. Integrating SF on the fibre walls reduced the pHRR by more than 45% compared with other fibre types, even though the temperature level was slightly higher. Removing the lignin from the outer cell wall of the fibre, as well as the amorphous components of hemicellulose between the fibres, reduces the overall thermal stability of KFs compared to raw cellulose matter [44]. Increasing the fibre’s thermal stability via surface modification can reduce the thermal decomposition rate as well as minimizing the HRC of combustion. The successful integration of pyrolyzed fibres within composite materials is dependent on the temperature applied to the fibres during pyrolysis. KFs are subjected to prior thermal treatment at approximate temperatures of 170 °C via caustic soda pulping. This process delignifies the fibres from the amorphous components, which can cause the morphology of the fibre walls to increase in surface roughness [45,46]. 

### 3.4. Mechanical Properties

#### 3.4.1. Compressive Strength

Figure 6 depicts the 7-, 14- and 28-day compressive strength. The standard deviation is shown via the error bars in Figure 6. The compressive strength is generally expected to decrease with fibre integration [47]. However, fibre composition within a modified cementitious matrix has been found to be a key factor in increasing the overall compressive strength of the KF composite. This is shown with SFKF105 having a compressive strength of 21 MPa at the 28-day interval. There was a 40% compressive strength increase after the 7- and 14-day intervals. A similar trend is shown with the control. At the 7- and 14-day intervals, the control had a compressive strength of 17 MPa, then a 47% strength increase to 25 MPa at the 28-day interval. Samples SFKF10 and KF10 had a moderate linear incline. Sample SFKF10 had 10, 13 and 14 MPa at the 7-, 14- and 28-day intervals, respectively, whereas KF10 had 8, 9 and 11 MPa at the 7-, 14- and 28-day intervals, respectively. Sample KF10 demonstrated that fibre modification is critical to increase the compressive strength of the cementitious composite. Furthermore, SFKF105 highlights the importance of modifying the matrix. Previously published reports [14,15,48] agree that the compressive strength decreases when fibre integration increases. This is mainly caused via voids and induced pockets within the composite formation. Due to this factor, fibre composites contain a reduced overall density. Therefore, increasing a composite’s density can also increase the compressive strength. During the heat of hydration, there is an exothermic reaction within the specimen’s matrix. This is specifically related to the reaction of water with pozzolanic materials within concrete, mainly cement [49]. This heat release entails the replacement of weak bonds with stronger ones. When the pozzolanic composition of the fibre cement composites is altered with MK, the exothermic reaction is altered, increasing the chemical bonding of the fibre within the matrix. Figure 6 illustrates the increased compressive strength when also modifying the constituent materials. 

#### 3.4.2. Flexural Strength

Figure 7 depicts the 7-, 14- and 28-day flexural strength. The modulus of rupture (MOR) of the composite specimens is depicted with the maximum flexural strength. The standard deviation is shown via the error bars in Figure 7. The control exhibited the highest strengths of 2.4, 2.4 and 2.6 MPa at 7-, 14- and 28-day intervals, respectively. Samples SFKF10, KF10 and SFKF105 withhold the same flexural strength, 1.5 MPa, at the 7-day interval. However, SFKF105 then had a 17% and 25% increase at the 14- and 28-day intervals with 1.8 MPa and 2 MPa, respectively. The integration of MK has been shown to increase the elastic properties of composite materials [50]. This is primarily due to the improvement of uniformity in the matrix. Moreover, the variation of viscosities and alkaline silicates of different cations can also affect the geopolymerization process. This in turn can create variations of the microstructure and influence the mechanical characteristics. It is important to note that raw KFs contained a higher MOR than modified fibres at the 28-day interval. Increasing the fibre content of the composite can also increase the flexural strength due to the crack bridging effect [48,51]. However, previous research showed an increased MOR when only 5% fibre integration occurs. With 5% fibre integration, there was a 25% flexural strength increase of raw KFs compared to SFKFs [15]. In this study, there was a 20% flexural strength increase of raw KFs compared to SFKFs at the 28-day interval. This demonstrated that modifying the fibre with SF can increase the fibre’s rigidity under flexural loading. Figure 8 illustrates SEM images of fibres within the concrete composite, with Figure 8A demonstrating fibre pull-out. This occurs when increased axial loading causes the fibre to pull out of the matrix, demonstrating a lack of mechanical and chemical bonding. The portion (see Figure 8B) shows the fibre wall unaffected by the cementitious matrix. Researchers [52] have shown that the primary fault of using natural fibres in cementitious composites is due to fibre deterioration. SF applied to the fibre walls appears to have reduced the deterioration. Figure 8C,D demonstrate the fibre embedded within the concrete composite. Furthermore, Figure 8D shows snapping of the fibre tip. This could be due to excessive loading previously applied to the composite. Generally, fibre composites can fail due to either fibre breakage or fibre pull-out when the load exceeds the composition’s tensional, compressional or shear forces [53]. 

#### 3.4.3. Tensile Strength

Figure 9 depicts the 7-, 14- and 28-day tensile strength. The coefficient variation of the tensile strength was between 0–0.2 MPa. The standard deviation is shown via the error bars in Figure 9. The fibre composition within a modified cementitious matrix is a key factor towards the overall tensile-strength-bearing capacity. Sample SFKF105 withheld values of 2.1, 2.6 and 2.6 MPa at 7, 14 and 28-day intervals, respectively. It is important to note that the maximum strength at day 28 was higher than the control. This signifies that reducing OPC via matrix modification does not affect the tensile-bearing capacity. Sample KF10 withheld the lowest tensile strength of 1.1, 1.2 and 1.6 MPa at 7-, 14- and 28-day intervals, respectively, although it is important to note that there was a similar linear incline as seen with SFKF10. Sample SFKF10 had moderate tensile strength of 1.7, 1.8 and 2 MPa at 7-, 14- and 28-day intervals, respectively. However, SFKF10 had the highest strength at day 28, which was the weakest for both the control and SFKF105 at their 7-day interval. This further demonstrates that although fibre modification can benefit the tensile strength of the composite, matrix modification is also required to enhance the composite’s strength. A previously published work [22] agrees that cementitious-based materials are tension weak and can easily form micro-cracks on the surface. As discussed, modifying the pozzolanic materials within cementitious-based composites can positively affect the exothermic reaction of the material’s microstructure. The replacement of weaker bonds with stronger bonds during this phase also enhances the correlation between the constituent materials of the composite. This is shown in Figure 9, with the increased tensile strength of the bespoke mix design.

## 4. Conclusions and Suggestions for Future Research

In this study, thermal and mechanical evaluations of fibres derived from cardboard waste within cementitious composites were conducted. Research highlighted the abundance of cardboard waste and a novel approach was defined to reduce potential accumulation in landfill areas, promoting a sustainable approach in construction. SF was implemented as the fibre surface modification technique and MK was used as the matrix modifier to enhance fibre durability. Cardboard waste was subjected to mechanical and thermal processing methods. These processes created KFs, ready for composite integration. Mechanical properties were determined via compressive, tensile and flexural testing methods. Morphology of the fibres was established using an SEM with an EDS provision. FT-IR was employed to illustrate the chemical nature of the materials. Thermal, calorimetric and combustion attributes of the fibres were measured using a TGA, DSC and PCFC, respectively. The SEM/EDS reports illustrated the successful application of SF onto the KF walls. This factor of surface modification using SF exhibited a reduction in the HRC when comparatively analysed with raw KFs, cardboard and cardboard pulp. This also gave SFKFs a relatively lower level of combustibility, which can ultimately lead to an improved durability of the fibres within cementitious composites when exposed to elevated temperatures/fires. The TGA results also favourably compared with the corresponding data obtained from the PCFC runs, especially in terms of the char residues obtained. DSC curves showed a broader endothermic peak beyond 400 °C, indicating a slow pyrolysis reaction following the main chain degradation. The FT-IR spectrum showed signals that are typical of ligno-cellulosic material with a character fingerprint region. The microstructure data revealed that the chemical processing applied to produce KFs does not affect the thermal composition of the cellulose matter. The integration of 5% MK within the cementitious matrix enhanced the compressive, tensile and flexural properties compared to non-modified matrixes. However, compressive and flexural strength was reduced when fibre integration occurred. Tensile strength properties were greater than the control when MK and SFKFs were integrated in the composite. This demonstrated the durability enhancement of surface modification using SF. Future research can be directed towards the fire performance of KF concrete and the reaction of isothermal conditions on the fibre durability. The major findings of this research are summarized as follows:Dried cardboard pulp exhibited a higher combustibility rate than its cardboard and fibrous counterpart; Inorganic materials such as SF can reduce the overall flammability of natural fibrous materials;A 10% KF integration exhibited satisfactory strength results for non-structural concrete; Waste cardboard can be utilized within cementitious composites; Matrix modification using MK enhances the mechanical strength of fibre-reinforced concrete; Using KFs derived from waste cardboard can reduce virgin resources requirement and divert waste in landfill management systems.

## Figures and Tables

**Figure 1 materials-15-08964-f001:**
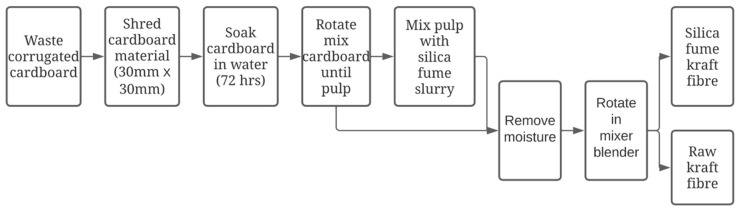
Fibre preparation methodology.

**Figure 2 materials-15-08964-f002:**
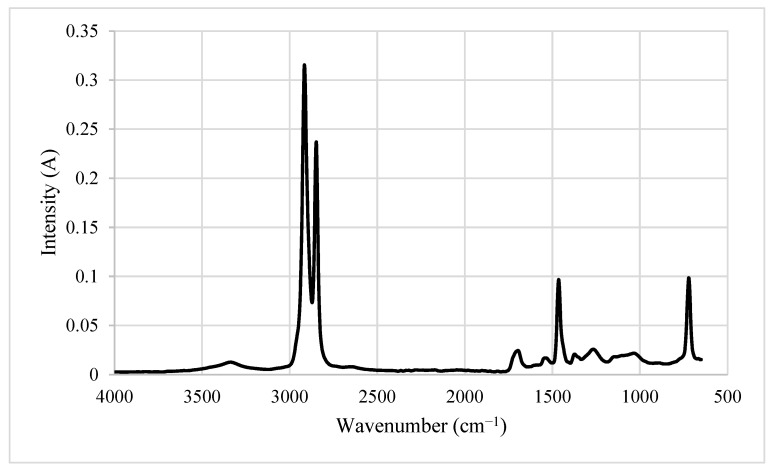
FT-IR spectrum of the dry pulp from cardboard (CBDP), where wavelength (in wave numbers: cm^−1^) are plotted against signal intensity (in arbitrary units).

**Figure 3 materials-15-08964-f003:**
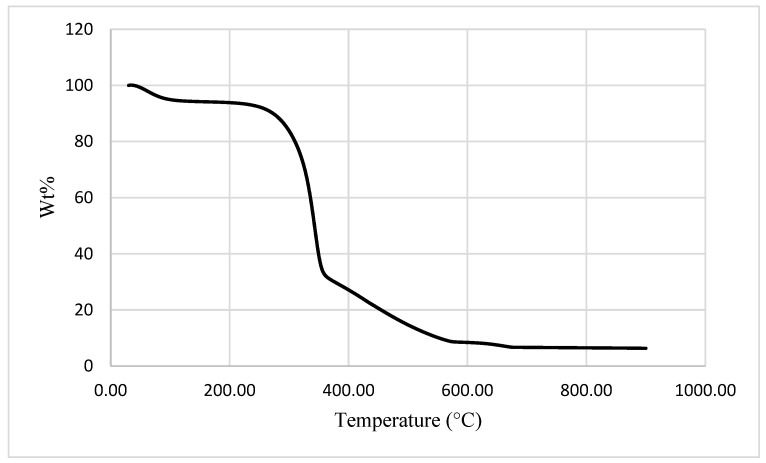
Thermogram obtained for CB sample at a heating rate of 10 °C min^−1^ in nitrogen.

**Figure 4 materials-15-08964-f004:**
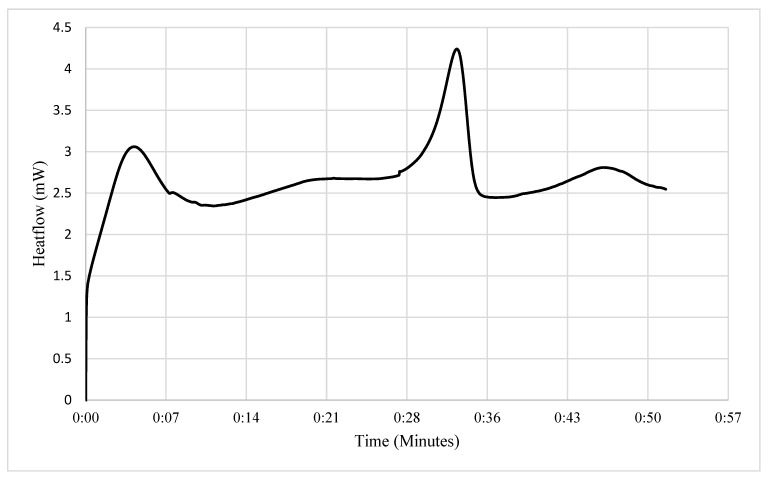
DSC curve of SFKFs showing three different endothermic processes.

**Figure 5 materials-15-08964-f005:**
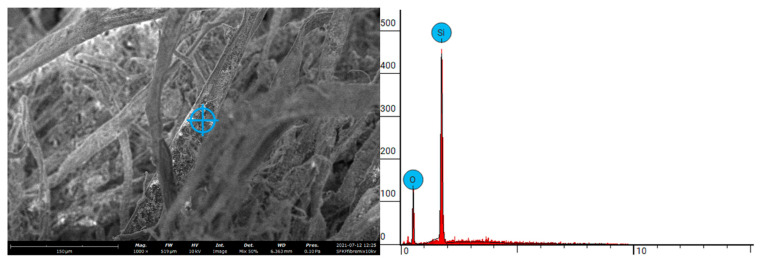
SEM/EDS report of SF on fibre.

**Figure 6 materials-15-08964-f006:**
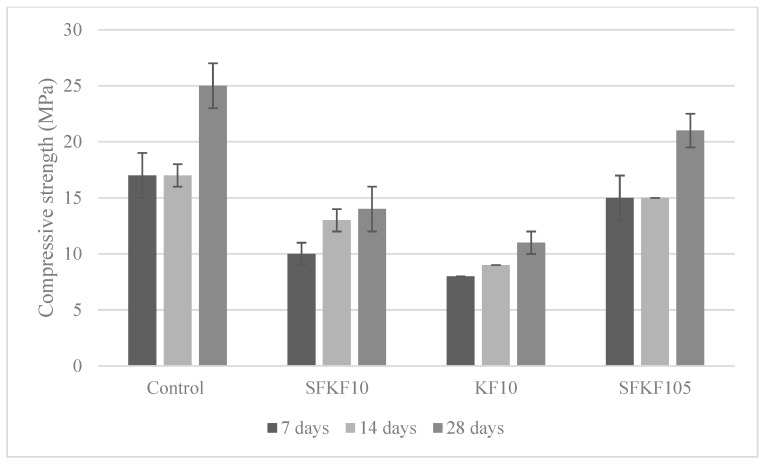
Variation of compressive strength.

**Figure 7 materials-15-08964-f007:**
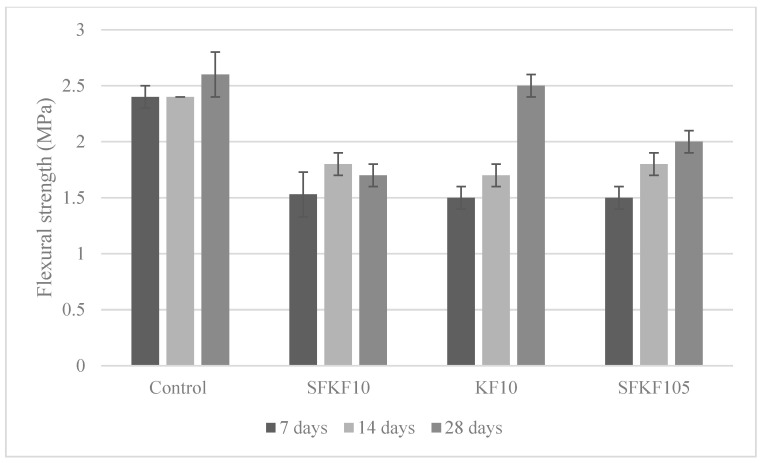
Variation of flexural strength.

**Figure 8 materials-15-08964-f008:**
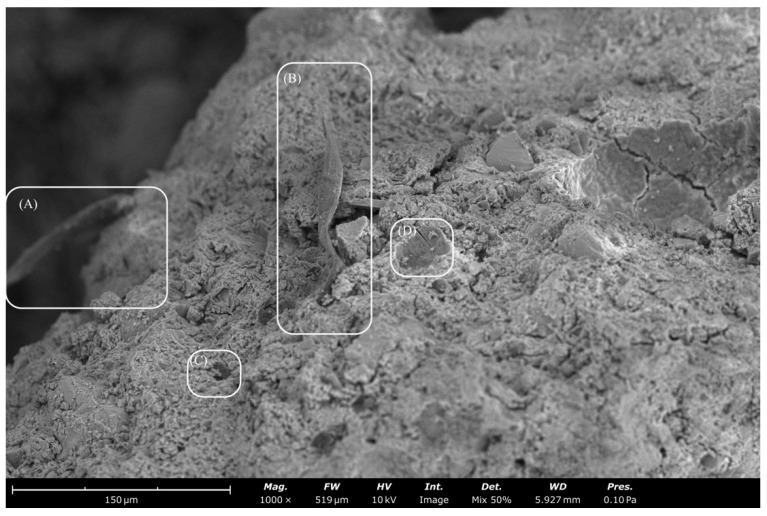
SEM images of fibres within concrete composite. (**A**,**B**) Retained fibre integrity, (**C**,**D**) Fibre snapping occurred.

**Figure 9 materials-15-08964-f009:**
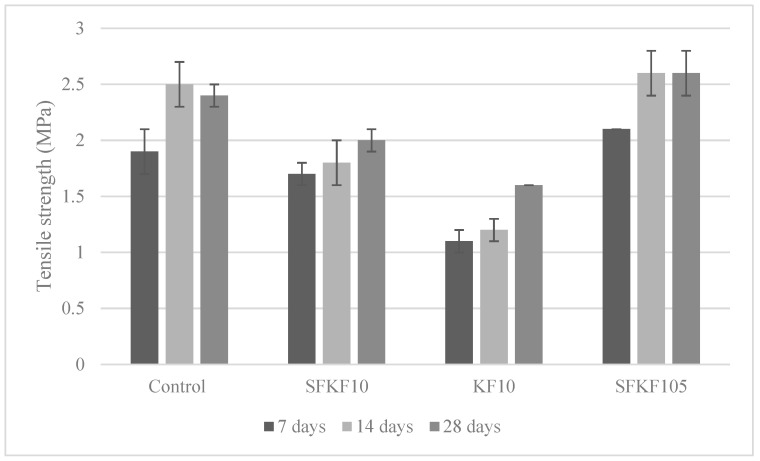
Variation of tensile strength.

**Table 1 materials-15-08964-t001:** Composition of materials adopted from [15].

Chemical	Material Component wt. %		
	MK	SF	OPC
SiO_2_	54–56	≥75–<100	19–23
Al_2_O_3_	40–42		2.5–6
Fe_2_O_3_	<1.4		
TiO_2_	<3.0		
SO_4_	<0.05		
P_2_O_5_	<0.2		
CaO	<0.1		61–67
MgO	<0.1		
Na_2_O	<0.05		
K_2_O	<0.4		
Amorphous silica, fumes		≥0.3–<1	
CaSO_4_·2H_2_O			3–8
CaCO_3_			0–7.5
Fe_2_O_3_			0–6
SO_3_			1.5–4.5

**Table 2 materials-15-08964-t002:** Composite constituent materials.

Mix Code	MK	KF	SFKF	OPC
Control	-	-	-	100
SFKF10	-	-	10	90
KF10	-	10	-	90
SFKF105	5	-	10	85

**Table 3 materials-15-08964-t003:** Mass losses and the amount of residue obtained during the TGA tests.

Sl. No.	Sample (Heating Rate: °C min^−1^)	Mass Loss at 180 °C (wt. %)	Mass Loss at 480 °C (wt. %)	Char Residue at 900 °C (wt. %)
1	CB (10)	7	94	2.4
2	CB (60)	5	77	16
3	CBDP (10)	6	83	6.0
4	CBDP (60)	4	76	16
5	KF (10)	3	72	11
6	KF (60)	4	72	20
7	SFKF (10)	4	59	31
8	SFKF (60)	3	49	44

**Table 4 materials-15-08964-t004:** The relevant parameters from the PCFC runs.

Sl. No.	Samples	Temp to pHRR (°C)	pHRR (W g^−1^)	THR (kJ g^−1^)	HRC (J g^−1^ K^−1^)	Char Yield (wt. %)	EHC (kJ g^−1^)
1	CB	365	133	9.57	132	17.3	11.6
2	CBDP	367	164	10.1	163	15.0	11.9
3	KF	368	139	9.30	139	22.1	12.4
4	SFKF	371	91.9	6.10	92	47.5	11.5

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
