# Peer review of "Thermal Characterizations of Waste Cardboard Kraft Fibres in the Context of Their Use as a Partial Cement Substitute within Concrete Composites"

_materials, 2022, doi:10.3390/ma15248964_

Round 1

Reviewer 1 Report

The submitted manuscript is about evaluation the thermal characterization of cardboard, cardboard pulp, raw and SF surface modified KFs. The presented manuscript seems to be interesting for readers of the Materials journal, it is written in a good manner and suits the requirements of the journal. It can be accepted for publication after recommended major revision. I kindly ask authors to prepare a response letter point-by-point rebuttal and must be subjected to the manuscript as well, considering the following comments with sufficient explanations.

1)     In the introduction, 2 different types of kaolinite and disordered kaolinite and their dehydroxylation (DHX) resulting productions of metakaolinite and metadiskaolinite at different temperatures through experimental and numerical investigations (DFT computational methods) must be reported, the following literature is used for the extraction of these information: DOI: 10.1007/s42860-020-00082-w. 

2)     I am wondering which type of MK (metakaolinite and metadiskaolinite) has been used for this study according to the following literature (DOI: 10.1007/s42860-020-00082-w)? It is highly suggested to use the XRD or NMR measurement for the samples, which have been used for this study to find out about the structure of stacked layers of MK. The reason is that the dissolution rate of Al and Si monomer are different for metakaolinte or metadiskaolinte or partially dehydroxylated metakaolinte and metadiskaolinte depending on the folded Al.

3)     Different participation of MK (metakaolinite and metadiskaolinite) can also affect the elastic constants properties, how can you explain it? 

4)      In Figure 2, the “wavenumber (cm-1)” must be written in the horizontal axis. 

5)     The symbol of temperature must be shown above the number, and it must be modified in the different places of the literature like lines 228, 229, 232, 237, 242, 252, …….   

6)     After number of each Figure in the caption, point must be implemented. Like “ Figure X. “

7)    Please make more comprehensive conclusion as in the revised version the following points must be included; materials and methods, the significant of this study, the scope of the effort, the procedures used to execute the work, and the major findings.

Best regards,

Author Response

Thank you for the valuable comments. The authors have significantly made revisions to improve the manuscript based reviewer's constructive criticism. 

The attached file provides a detailed expalanation of the responses to changes. 

Reviewer 2 Report

Dear Authority,

The manuscript entitled ‘Thermal characterizations of waste cardboard kraft fibres in the  context of their use as a partial cement substitute within concrete composites’ reveals the structural characterization of Kraft Fibre and Silica Fume Kraft Fibre reinforced into concretes as a partial substitute for cement . I think, the paper presents valuable information about variation in both thermal, calorimetric and combustion behaviour of the fibres in concerete along with some mechanical properties of composites such as tensile, compression and flexural strength. Overall, the data in manuscript is well represented and gives important information on Silica modified composites (SFKF) contributing lower heat release capacity (HRC).   However, there is missing information in abstract about mechanical properties. The manuscript needs to be revisited by considering following comments;

 Abstract is not good enough, improve it with the numerical results and write it again succinctly. Abstract section should be concisely reflected the content and summarize the problem, the method, the results, and the conclusions. Also, please mention about mechanical properties of SFKF reinforced composite system in terms of tensile, compression and flexural strength.

After minor modification, the paper could be considered for publication in Materials.

Best wishes,

Author Response

Thank you for the comments raised by the reviewer #2. 

The authors have carefully considered the comments and revised the manuscript. The attached provides a detailed expalantion of the changes made in the revised manuscript. 

Round 2

Reviewer 1 Report

Dear Authors,

thanks and it is accepted in present form.

Best regards,